# Tracking cells in epithelial acini by light sheet microscopy reveals proximity effects in breast cancer initiation

Ashna Alladin[1†], Lucas Chaible[1†], Lucia Garcia del Valle[1], Reither Sabine[2], Monika Loeschinger[3], Malte Wachsmuth[3], Jean-Karim Hériché[1], Christian Tischer[4], Martin Jechlinger[1*]

[1]Cell Biology and Biophysics Unit, EMBL, Heidelberg, Germany; [2]Advanced Light Microscopy Facility, EMBL, Heidelberg, Germany; [3]Luxendo Light-Sheet, Bruker Corporation, Heidelberg, Germany; [4]Centre for Bioimage Analysis, EMBL, Heidelberg, Germany

**Abstract** Cancer clone evolution takes place within tissue ecosystem habitats. But, how exactly tumors arise from a few malignant cells within an intact epithelium is a central, yet unanswered question. This is mainly due to the inaccessibility of this process to longitudinal imaging together with a lack of systems that model the progression of a fraction of transformed cells within a tissue. Here, we developed a new methodology based on primary mouse mammary epithelial acini, where oncogenes can be switched on in single cells within an otherwise normal epithelial cell layer. We combine this stochastic breast tumor induction model with inverted light-sheet imaging to study single-cell behavior for up to four days and analyze cell fates utilizing a newly developed image-data analysis workflow. The power of this integrated approach is illustrated by us finding that small local clusters of transformed cells form tumors while isolated transformed cells do not.

*For correspondence:
martin.jechlinger@embl.de

†These authors contributed equally to this work

## Introduction

Organoid cultures grown from cell-lines or primary cells have been successfully employed to study molecular mechanisms during different stages of tumorigenesis (*Clevers, 2016*; *Havas et al., 2017*; *Simian and Bissell, 2017*). However, they are derived from primary material that usually allows oncogenic activation in all cells of the tissue and therefore cannot reproduce the localized expansion at a defined part of the tissue that is seen in the patient situation. To study tumor initiation, models need to be established wherein only a few tumorigenic cells expand in the context of its immediate non-tumor microenvironment (*Tabassum and Polyak, 2015*) and the interplay of these populations can be visualized.

Long-term imaging of primary organoids has been achieved via light-sheet microscopy (*Drost et al., 2015*; *Serra et al., 2019*; *Verissimo et al., 2016*) and these efforts have benefited from the lower phototoxicity provided by SPIM imaging. However, regarding fast dividing cells in tumor organoids, tracking single-cell dynamics necessitates high resolution imaging which in turn limits the time frame in which organoids can be imaged without phototoxic effects (*Held et al., 2018*). Conversely, imaging primary organoids for longer time periods requires an offset of temporal and cellular resolution that eventually cannot allow single-cell fate tracking (*Dekkers et al., 2016*).

Two models have been used to understand tumor progression and heterogeneity (*Plaks et al., 2015*); the (i) hierarchical model refers to tumor propagating cells as cancer stem cells (CSCs) (*Kreso and Dick, 2014*) and the (ii) stochastic model states that every cell within a tumor is equally likely to be the cell of origin and facilitate tumor initiation and progression (*Vogelstein et al., 1988*; *Williams et al., 2007*). These models for tumor propagation can be reconciled when considering

**eLife digest** There are now drugs to treat many types of cancer, but questions still remain around how these diseases start in the first place. Researchers think that tumor growth begins when a single cell suffers damage to certain sites in its DNA that eventually cause it to divide uncontrollably. That damaged cell, and its descendants, go on to form a lump, or tumor. The trouble with proving this theory is that it is hard to watch it happening in real time. Doctors usually only meet people with cancer when their tumors start to cause health problems. By this point, the tumors contain millions of cells.

A way to watch the very beginnings of a cancer could reveal risk factors within a tissue that foster the growth of a tumor. But first, researchers need to test their theory about how the disease begins in the first place. One way to do this is to surround a single cancer cell with healthy cells and watch what happens next. To do this, Alladin, Chaible et al. took healthy cells from the breast tissue of mice and grew them in the laboratory into mini-organs called organoids. These organoids share a lot of features with actual mouse breast tissue; they can even make milk if given the right hormones.

Once the organoids were ready, Alladin, Chaible et al then started modifying a small number of single cells inside them by switching on genes called oncogenes, which are known to drive cancer formation in humans. Using fluorescent proteins and a sheet of laser light it was possible to watch what happened to the cells over time. This revealed that, even though all the oncogene-driven single cells received the same signals, not all of them started to divide uncontrollably. In fact, a single modified cell had a low chance of forming a tumor on its own. The more oncogene-driven cells there were near to each other, the more likely they were to form tumors. Alladin, Chaible et al. think that this is because the healthy tissue interacts with the modified, oncogene-driven cells to suppress tumor formation. It is only when a larger number of modified cells group together and start to communicate with each other that they can override the inhibitory messages of the healthy tissue.

How healthy tissue stops single modified cells from forming tumors is not yet clear. But, with this new mini-organ system, researchers now have the tools to investigate. In the future, this could lead to new strategies to stop cancer before it has a chance to get started.

that the population of oncogene driven cells have the capacity to interconvert between differentiated- and stem-like states (*Chaffer and Weinberg, 2015*; *Gupta et al., 2011*; *Plaks et al., 2015*), a flexibility that is supported by the microenvironment (*McGovern et al., 2009*; *Quail et al., 2012*). These concepts, however, have not been probed in the context of tumor initiation upon stochastic oncogene activation in an established cell-layer.

Here we present a novel model of breast tumorigenesis where only single cells express oncogenes within healthy mammary acini and are able to establish localized outgrowth. We thereby overcome the above-mentioned limitations of studying tumorigenesis events in the face of tissue-wide transformation. Furthermore, we perform long-term imaging of these stochastic breast tumor acini for the first time at a temporal resolution that allows us to follow single-cell fates. We also integrate this approach with an image analysis pipeline capable of segmenting cells in their dynamic progression towards tumorigenesis, so they can be tracked individually over time.

## Results

For modeling tumorigenesis in breast tissue, we use an inducible mouse model of breast cancer (TetO-MYC/TetO-Neu/MMTV-rtTA transgenic mice) (*Fry et al., 2016*; *Moody et al., 2002*; *Podsypanina et al., 2008*) that has been shown to recapitulate hallmarks of human breast disease (*Havas et al., 2017*; *Jechlinger et al., 2009*; *Figure 1a*). In this tractable transgenic mouse model, the activity of two potent oncogenes – *MYC* and *Neu* (the rodent homolog for the human *ERBB2* gene used in this mouse model)– are under the control of a Tet-O promoter that can only be activated in the presence of *both* the rtTA (reverse tetracycline-controlled transactivator) protein and the doxycycline compound. This three-part inducible system allows for temporal control of oncogenic expression by regulating doxycycline in the medium or animal diet and spatial control by the MMTV promoter that confines expression of the rtTA protein to the cells of the mammary lineage

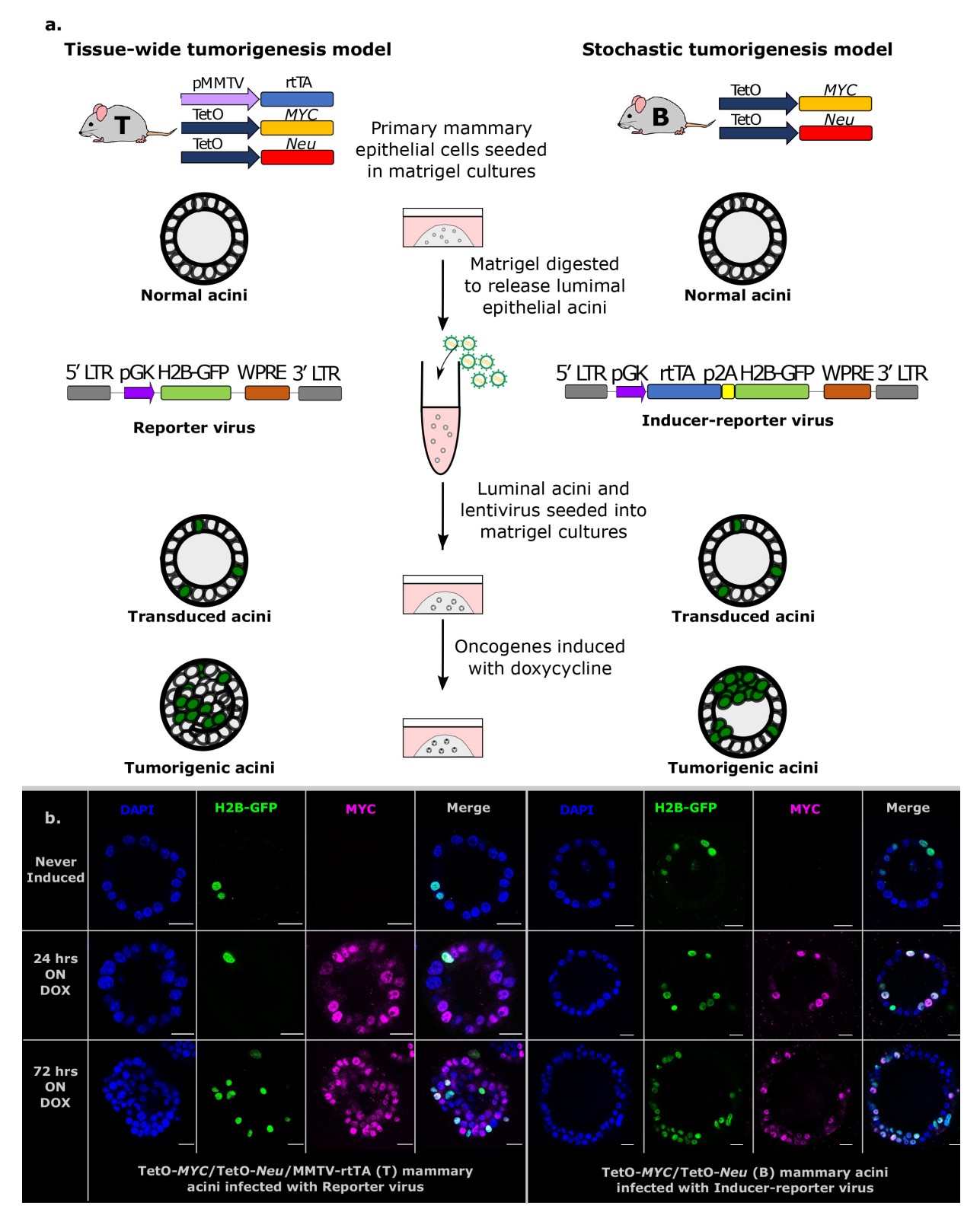

**Figure 1.** Establishment of stochastic tumorigenesis in mammary acini. (**a**) Schematic representation of the mouse models and the in vitro culture methods used. Acini are grown from single cells harvested from the mammary glands of either bi-trangenic (B) or tri-trangenic (T) mice, transduced with lentiviral particles in solution and re-seeded into 3D cultures. Doxycycline is added to the medium to induce the expression of oncogenes in cells expressing rtTA. B mice have the *MYC* and *Neu* oncogene constructs in their genome. These oncogenes are activated in single cells infected with the

*Figure 1 continued on next page*

Figure 1 continued

Inducer-reporter (pLenti-rtTA-GFP) lentiviral particles in the presence of doxycycline, modeling stochastic breast tumorigenesis (right panel). T mice have the rtTA transducer construct along with the oncogenes and all cells in T acini can be induced to express oncogenes in 3D culture in the presence of doxycycline. T mice infected with Reporter (pLenti-NULL-GFP) lentiviral particles are used as infection controls (left panel). Both viral particles mark single cells in the acini with H2B-GFP. (b) Representative immunofluorescence staining images of fixed 3D gels with B acini transduced with Inducer-reporter virus or T acini transduced with Reporter virus before induction (top), 24 hr post induction and (middle) and 72 hr post induction (bottom) with doxycycline. GFP expressing transduced cells (green), MYC oncoprotein (magenta), DAPI nuclear stain (blue). Scale bar, 10 µm.

The online version of this article includes the following figure supplement(s) for figure 1:

**Figure supplement 1.** Normalization of doxycycline concentration for the stochastic model using qPCR analysis.

(*Bockamp et al., 2002*). We modify this tissue wide tumorigenesis model (tri-transgenic (T) model; *Figure 1a*, upper panels) to generate a stochastic system by retaining only the oncogenic constructs, making them TetO-MYC/TetO-Neu mice (bi-transgenic (B) model; *Figure 1a*, lower panels). The rtTA inducer gene is then lentivirally delivered to single cells within luminal epithelial acini derived from the bi-transgenic (B) mouse glands, granting only this subset of cells the competence to express transforming oncogenes and thereby preventing tissue wide transformation. Hence, this novel (B) model should allow acini to present with two cell populations –tumorigenic and non-tumorigenic– and could be used to study the dynamics of early tumorigenesis in an otherwise normal epithelium (*Figure 1a*, lower right panels).

Primary mammary epithelial cells derived from transgenic mice were seeded in 3D matrigel as single cells to be grown for 48 hr to form small acini consisting of around 15 cells that arrange around a small lumen. A small number of single cells in these acini were then transduced with lentiviral particles (*Figure 1a*, middle panel). In the tissue-wide tumorigenesis model(T), acini were transduced with the reporter virus (pLv-pGK-H2B-GFP) that marks a subset of cells with H2B-GFP, while tissue wide rtTA expression is driven in all cells (*Figure 1a*, left panels). To achieve stochastic tumorigenesis, bi-transgenic (B) acini were transduced with the inducer-reporter virus (pLv-pGK-rtTA-p2A-H2B-GFP) that expresses rtTA and reporter H2B-GFP in only these single cells within the normal epithelium (*Figure 1a*, right panels). Then, doxycycline was supplemented in the medium to induce tumorigenic growth in rtTA expressing cells. Immunofluorescent staining of 3D matrigel cultures, for both sets of doxycycline-induced transduced acini, was used to validate transgene specific protein expression of the *MYC* oncogene (human) in only the transduced cells of (B) acini as opposed to all cells of (T) acini (*Figure 1b*, anti-human MYC specific antibody). We FACS sorted H2B-GFP marked cells from both systems respectively for qPCR analysis of transgene specific *MYC* and *Neu* mRNA expression. This way we normalized doxycycline dosage in both systems to obtain the same mRNA expression levels of oncogenes using 800 ng/ml doxycycline in the tissue wide (T) system and 600 ng/ml doxycycline in the stochastic (B) system (*Figure 1—figure supplement 1*).

To verify oncogene induction patterns and to understand the impact on epithelial morphology in the acini, 3D matrigel cultures of both doxycycline-induced (T) and (B) transduced acini were analyzed by immunofluorescent staining for polarity marker distribution. Lentivirus-transduced structures from the tissue wide induction system (T) presented with filled lumens that contained both infected (GFP positive) as well as non-infected oncogene-expressing cells. These structures exhibited overall disrupted epithelial polarity and some random microlumen, as reported previously (*Jechlinger et al., 2009*; *Figure 2a*). For the stochastic tumor induction system (B), we observed 2 phenotypes after 96 hr of oncogene expression; some of the (B) acini showed exclusively H2B-GFP marked cells in hyperplastic areas that also displayed double cell rim morphology and disturbed epithelial polarity (*Figures 2a* and 96 hours ON DOX, left panels), while the other (B) acini did not show expansion of the few H2B-GFP positive cells (*Figures 2a* and 96 hours ON DOX, right panels). Intrigued by these 2 distinct phenotypes, we analyzed transduced (H2B-GFP expression) (B) acini for their oncogenic MYC expression (*Figure 2b*). We verified that multi layered regions in acini contained mainly H2B-GFP positive cells that expectedly stained for MYC (*Figure 2b*, middle panels). For normal appearing regions, this analysis showed that the inability of single H2B-GFP positive, transduced cells to expand over 96 hr was not due to the lack of MYC expression (*Figure 2b*, lower panels).

To follow up these observations in more detail over time, we bred the nuclear reporter H2B-mCherry into the T and B mice to mark all the cells in the acinus for inverted light-sheet microscopy

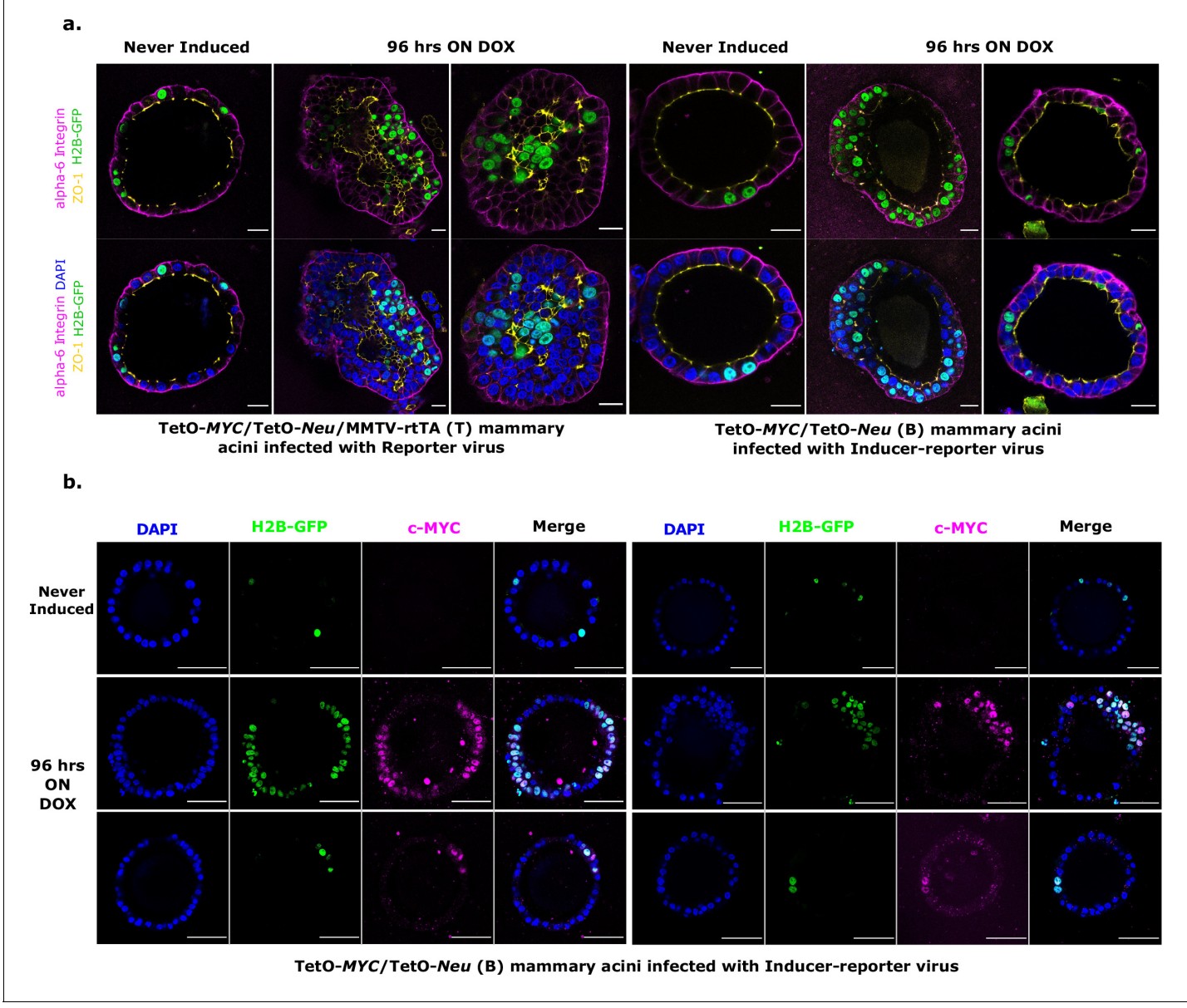

**Figure 2.** Characterization of stochastic tumorigenesis in mammary acini. (**a**) Representative immunofluorescence staining images of fixed 3D gels with T acini transduced with Reporter virus (left panels) and B acini transduced with Inducer-reporter virus (right panels), before induction and 96 hr post induction with doxycycline. Polarity markers include alpha-6-Integrin (magenta) and ZO-1 (yellow). Transduced cells are marked with GFP (green) and nucleus is counterstained with DAPI (blue). Scale bar, 20 μm.(**b**) Representative immunofluorescence staining images of fixed 3D gels with B acini transduced with Inducer-reporter virus before induction (top panels) and 96 hr post induction (middle and bottom panels) with doxycycline. GFP expressing transduced cells (green), MYC oncoprotein (magenta), DAPI nuclear stain (blue). Scale bar, 50 μm.

(Luxendo InVi SPIM, *Figure 3—figure supplement 1*). The InVI SPIM was adjusted for non-photo-toxic, long-term imaging (up to 4 days, every 10 min with 1 μm z-spacing). The T/H2B-mCherry acini transduced with reporter virus proliferated swiftly upon doxycycline addition showing expansion of both the marked and unmarked cells (*Figure 3a*); a sturdy tumor phenotype developed, manifested by multi-cell-layered rims and pronounced proliferation-associated-apoptosis in all acini (*Figure 3—Video 1*). In contrast, B/H2B-mCherry acini transduced with inducer-reporter virus, displayed pheno-typic variation upon induction of oncogenes in the transduced cells. Some acini showed fast clonal expansion of oncogene-expressing cells that form multilayer clusters in the acinus rim. This prolifer-ative phenotype seems to stem from several transduced cells in vicinity to each other at the start of

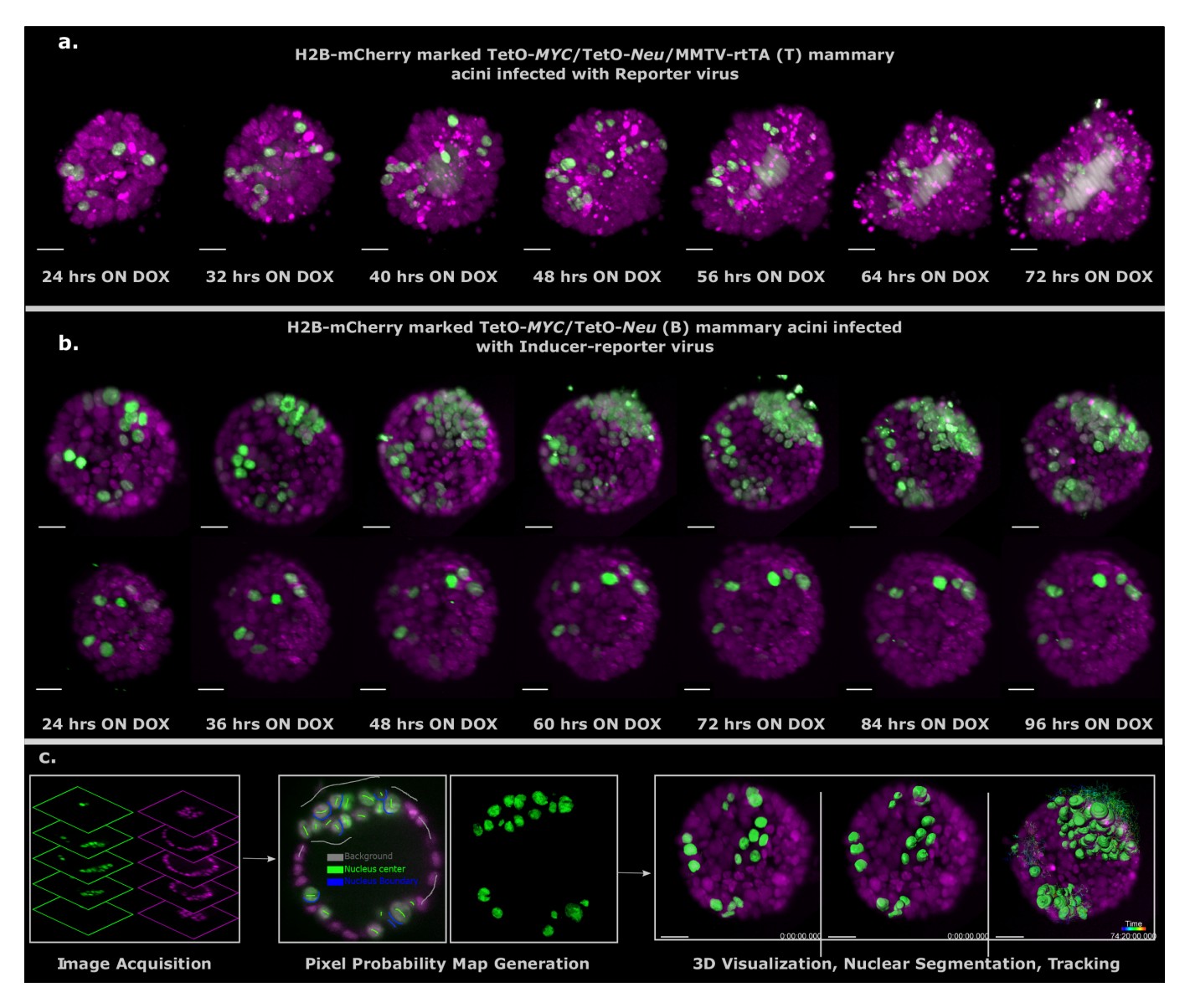

**Figure 3.** Light sheet imaging of stochastic tumorigenesis in mammary acini 3D images of selected timepoints during live-cell time-lapse microscopy of induced T acini transduced with Reporter virus. (a) or B acini transduced with Inducer-reporter virus (b). All cells in the acini express H2B-mCherry (magenta) and only cells transduced with lentiviral particles express H2B-GFP (green). Imaging was started 24 hr after oncogenic induction with doxycycline. In (b) the upper panel shows the proliferative phenotype seen with stochastic transformation, whereas the lower panel shows the non-proliferative phenotype observed in some stochastically transformed acini. (Imaging conditions: H2B-mCherry 594 nm Ex, 610 LP Em; H2B-GFP 488 nm Ex and 497–554 nm Em). Scale bar, 20 µm. (c) Schematic representation of the big-image data analysis pipeline developed to analyze the light sheet microscopy images. Images are acquired in two channels (H2B-mCherry in magenta and H2B-GFP in green) at 10 min intervals for 3–4 days. Big Data Processor Fiji plugin is used to pre-process the raw images and CATS Fiji plugin is used for generation of pixel probability maps (*Figure 3—figure supplement 2*). Image pixels of the H2B-GFP images are classified into background (black), nucleus center (green), nucleus boundary (blue) classes by manual training. Processed raw images along with the probability maps from the nucleus center channel (green) are exported to Imaris for 3D visualization, nuclear segmentation and single-cell tracking.

The online version of this article includes the following video and figure supplement(s) for figure 3:

**Figure supplement 1.** Sample holder preparation and sample mounting.
**Figure supplement 2.** Image pre-processing and pixel probability generation using Fiji plugins.
**Figure supplement 3.** Cell segmentation accuracy Image panels show the H2B-mCherry signal (magenta) along with the segmented H2B-GFP cells of an acinus at four equidistant timepoints.

*Figure 3 continued on next page*

*Figure 3 continued*

**Figure 3—video 1.** Video file for acinus shown in *Figure 3a*, showing fluorescence SPIM miscroscopy data in 2-color 3D projections (mcherry- magenta; GFP- green) over 72 hr of induction with doxycycline.

https://elifesciences.org/articles/54066#fig3video1

**Figure 3—video 2.** Video file for acinus shown in *Figure 3b* (upper panel), showing fluorescence SPIM miscroscopy data in 2-color 3D projections (mcherry- magenta; GFP- green) over 72 hr of induction with doxycycline.

https://elifesciences.org/articles/54066#fig3video2

**Figure 3—video 3.** Video file for acinus shown in *Figure 3b* (lower panel,, showing fluorescence SPIM miscroscopy data in 2-color 3D projections (mcherry- magenta; GFP- green) over 72 hr of induction with doxycycline.

https://elifesciences.org/articles/54066#fig3video3

time-lapse imaging (*Figure 3b*, upper panel; *Figure 3—Video 2*). Again, more sparsely infected acini, did not sustain proliferation of the oncogene-expressing cells (*Figure 3b*, lower panel; *Figure 3—Video 3*). Taken together, the phenotypes for (B) and (T) acini are consistent with the respective behavior observed with the 96 hr end point immunofluorescent analysis (*Figure 2a and b*), verifying these prior observations and excluding phototoxic effects due to longitudinal SPIM imaging.

To analyze the dual-color light-sheet movies (H2B-mCherry—all cells in the acinus , H2B-GFP— transduced cells within the acinus) on a single-cell level, we developed a big image data compatible image analysis pipeline that allows efficient visualization of longitudinal big image data, nuclear segmentation and single-cell tracking in 3D (*Figure 3c*, *Figure 3—figure supplement 3* and *Figure 4— figure supplement 2*). The Big Data Tools plugin (Big Data Processor (Tischer, Norlin, and Pepperkok)) for Fiji (*Schindelin et al., 2012*) was used to visualize the 2D image stacks recorded over time. It uses lazy loading to stream the image files, which can be up to a few terabytes in size per acinus over 3–4 days of imaging. It also allows for easy cropping, binning, chromatic shift correction between channels and file format conversion to Imaris compatible formats. Once the raw data have been pre-processed, the CATS plug-in from Fiji (Tischer and Pepperkok) was used for segmenting the cell nuclei in the green channel images. The CATS plug-in uses machine learning algorithms to predict the probabilities of all pixels in the image and classify them into pre-defined classes. Three classes — background, nucleus, boundary — were trained manually on the CATS tool and the segmented images were exported to the commercially available Imaris software (*Bitplane AG, 2020*, Software available at http://bitplane.com) for 3D rendering and tracking. The nucleus class channel from the CATS tool was used for surface rendering and the surfaces in all time points were tracked over the course of the movie. Cell tracking allowed us to follow the clonal evolution for each transduced cell in the acinus over 3 days. Single transduced cells within one acinus show a difference in proliferation and cell fate as indicated in representative tracks (*Figure 4a*).

To better understand the parameters that positively affect a transduced cell in the stochastic tumorigenesis model to start proliferating and establishing a tumor within a normal epithelium, we extracted the center of mass coordinates of all cells in the acinus at the start of the imaging and represented them as dots in a 3D space: depicted as magenta for normal cells and green for transduced cells in *Figure 4b* (upper right panel). Equipped with the tumor formation success of each stochastically transduced acinus, we hypothesized that tumors originate from groups of independently transduced oncogene-expressing cells in closer vicinity within the acinus. To explore this in detail, we defined clusters of cells in the stochastic model as a group of oncogene-expressing cells that are closer to each other than to other oncogene-expressing cells of the same acinus. Notably, these clusters contain both oncogene-expressing cells as well as normal cells of the acinus. The cluster volume was described as the sphere located at the center of mass of a cluster with a diameter equal to the distance between the two farthest oncogene-expressing cells of the cluster. Next, we extracted 9 features that represent reasonable factors that could influence tumor establishment in the epithelium such as size of the acinus (represented by number of cells in the acinus), density of cells in the acinus, infection efficiency (represented by number of infected cells in the acinus), number of cells in a defined cluster volume, number of independently transduced cells in a cluster volume, distances between all cells in a cluster volume, distances between transduced cells in a cluster volume, fraction of transduced cells in the cluster volume and number of cell-cell contacts between transduced cells in a cluster volume (*Figure 4b*, left panel).

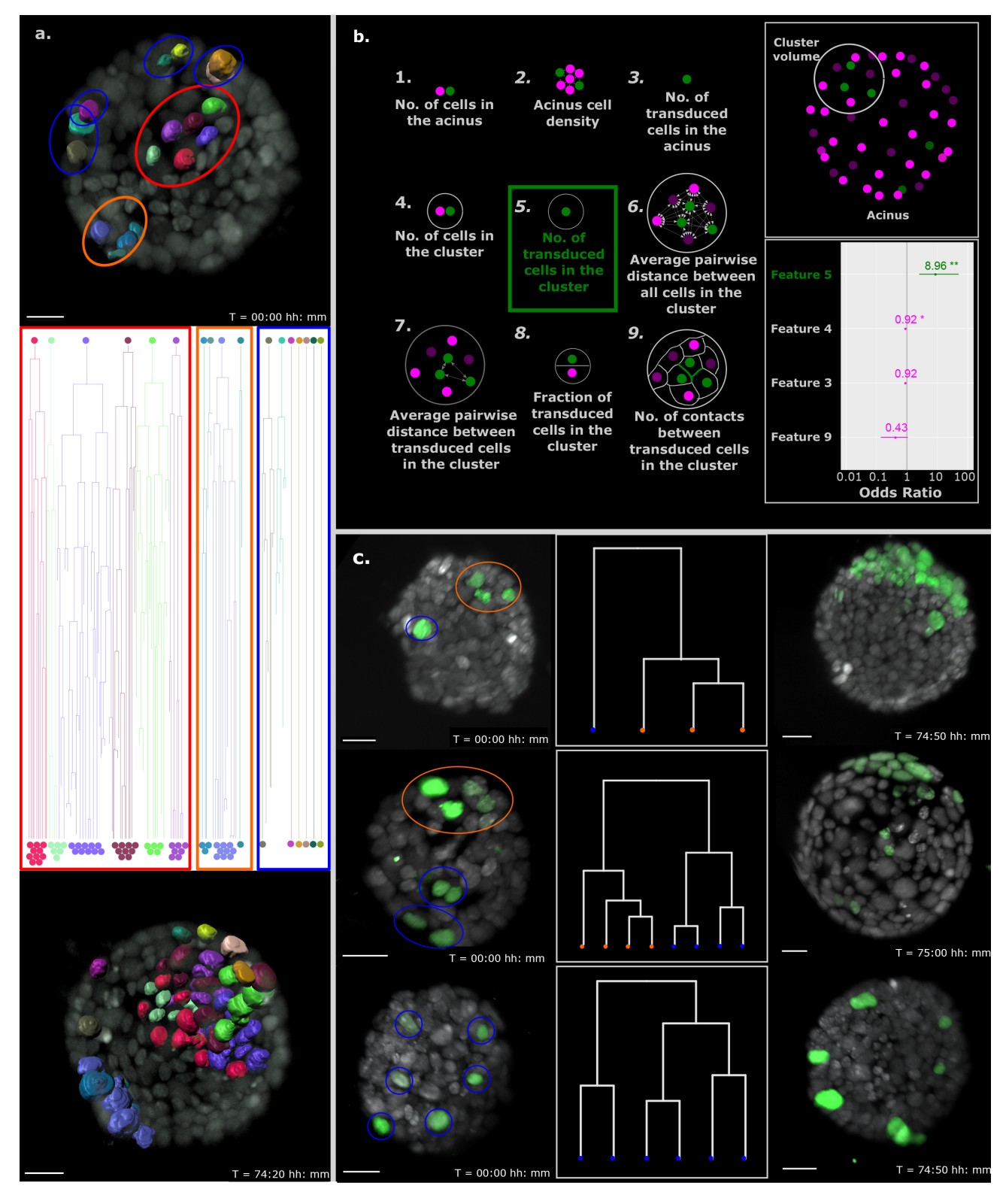

**Figure 4.** Proximity of transformed cells in a normal epithelium enhances tumor proliferation and establishment. (a) Single-cell tracking results for every cell in a representative B acinus transduced with the Inducer-reporter virus. Top panel shows the acinus at the beginning of the time-lapse (24 hr post induction) with each transduced cell surface rendered with Imaris. The middle panel shows the lineage trees of each individual cell over the time-lapse recording. Lineage trees of single cells are grouped into proliferative (highlighted in red, orange) and non-proliferative (highlighted in blue) cell

*Figure 4 continued on next page*

*Figure 4 continued*

clusters. The bottom panel shows the acinus at the end of the time-lapse (~76 hr post induction with doxycycline). Color coding of each cell maintained in all panels. Scale bar, 15 µm. (**b**) Schematic representation of the 9 features of stochastically transformed cells extracted at the beginning of time-lapse imaging. These features were assessed for their impact on tumor cell proliferation within B acini transduced with the Inducer-reporter virus using logistic regression. Lower right panel: Coefficients (represented as odds ratios) of the three features included in the best logistic regression model, colored horizontal bars represent the 95% confidence interval of the estimate. ** indicates p-value (of having no effect)<0.01, * indicates p-value<0.05. The vertical grey line indicates the position of no effect. (**c**) Representative B mammary acini stochastically transduced with the Inducer-reporter virus and induced with doxycycline. Left panels show acini 24 hr post induction. Color highlights indicate clusters of transduced cells identified from hierarchical clustering (shown in middle panels) with proliferative clusters highlighted in orange and non-proliferative clusters highlighted in blue. Right panels show the same acini ~ 72–76 hr post induction. Scale bar, 20 µm.

The online version of this article includes the following figure supplement(s) for figure 4:

**Figure supplement 1.** Acini feature analysis to exclude acinus-specific effects on proliferation of transduced cell clusters.

**Figure supplement 2.** Proliferative and non-proliferative cell clusters can be found within the same B acinus.

We analyzed 20 acini and all the transduced cells in these acini (n = 150) at the start of the imaging to ascertain the effect of the immediate non-transformed cell microenvironment on tumor cell proliferation. Since clusters of oncogene-expressing cells could be tracked over time, each cluster was associated with either a tumor outcome, or not. To identify which features were linked to this outcome, we chose a model selection approach, based on information theory. We fitted a logistic regression model for all possible linear combinations of features and selected the best model based on the Akaike information criterion (with correction for small sample sizes) (*Calcagno and Mazancourt, 2010*). In this model only one feature, the 'number of transduced cells in the cluster', can predict whether a cluster can form a tumor. Each additional transduced cell in a cluster increases the odds of this cluster forming a tumor by 9 (*Figure 4b*, lower right panel). Since other models within a few information criterion units of the best one can be considered equally good at predicting tumor formation, we also looked at features included in models within 3 information criterion units from the best one. Relative importance of features can be estimated by the sum of the information criterion values of the models in which each feature appears (*Buckland et al., 1997*). Computing relative importance across the selected models also revealed 'number of transduced cells in the cluster' as the most important feature, followed by 'number of cells in a cluster' and 'number of contacts between transduced cells in a cluster'. However, the only feature with significant contribution to tumor formation is 'number of transduced cells in the cluster'. It is important to note that this result does not rule out contribution from other factors which more targeted experiments may be able to reveal.

We also analyzed the effect of whole acinar features at the beginning and end of the imaging period to try and detect acinus-specific advantages, like acinus size, cell density and cell proliferative rate. Acini with transduced cell clusters that displayed a tumor outcome were compared to acini where the cell clusters did not result in tumor outcomes (*Figure 4—figure supplement 1a*). Acini with tumors only differed significantly from those without tumor on the number and proliferation rate of transduced cells in line with the observation that clusters of transduced cells are the primary indicators of tumor outcome. To account for this and possibly other acinus-specific effects, we also evaluated a logistic regression model including the three most important features identified above plus a random effect for acinus of origin and interaction terms for the starting number of normal cells in the acinus. In this model again, the number of oncogene-expressing cells in the cluster appeared as the main contributing factor to tumorigenesis with an odds ratio of 6.7 (*Figure 4—figure supplement 1b*).

That tumors are mainly initiated by clusters of transduced cells is further illustrated by representative acini shown in *Figure 4c*; proliferating cells that establish tumor foci at the end of the long-term imaging cluster together at the start of the time lapse imaging and the non-proliferative cells are more sparsely located within the acinus. In line, we found examples of similar sized acini bearing the same number of transduced cells that showed differential cluster expansion depending on the 'numbers of transduced cells within a cluster' (*Figure 4—figure supplement 2*; comparison a to b). Consequently, and again depending on the number of cells within the specific cluster, we observed clusters behaving differently within the same organoid; the organoids shown in *Figure 4a*, *Figure 4c* (top and middle panels) and *Figure 4—figure supplement 2a*, display clusters of single cells that do

not expand over time, whereas clusters containing several cells show massive expansion (beyond normal expected cells division rates due to growth of the acinus).

## Discussion

Our results indicate that a proximity-controlled interaction or signaling network between different transformed cells might guide tumor outgrowth in a normal epithelium (*Greaves and Maley, 2012*; *Reid et al., 2010*). This might be due to the repressive effect that an intact polarized tissue layer exerts on single early stage cancerous cells (*Lee and Vasioukhin, 2008*). According to our modeling attempts the simple parameter of 'number of transduced cells within a cluster' is enough to impinge on overall epithelial integrity and most likely already accounts for sub-features like 'distances between transduced cells in a cluster volume' or 'number of cell-cell contacts between transduced cells in a cluster volume', which did not pan out as predictive factors on their own. These observations, however, do not exclude the importance of such traits, which should be explored in more detail for the different situations we report (proliferative versus non proliferative clusters). In addition, the increased probability for tumor outgrowth upon proximity effects of closely located transformed cells, might also be rooted in paracrine effects (e.g. via microRNAs [*Kosaka et al., 2012*] in the cluster, or hydrogen ion dynamics/pH [*Reshkin et al., 2014*]). The questions outlined above can now be assessed employing the presented 3D system and longitudinal imaging pipeline together with appropriate fluorescent-tagged reporter proteins.

Studies on loss of important polarity proteins have highlighted their function as non-canonical tumor suppressors in breast tumorigenesis (*McCaffrey et al., 2012*; *Partanen et al., 2012*); however, other reports (*Macara and McCaffrey, 2013*; *Xue et al., 2013*) cannot confirm these observations which were all obtained from tissue wide transformed model systems, leaving conclusions open for alternative studies. Similarly, it will interesting to trace the effect of prominent tumor factors in breast cancer, such as *SCRIBL/MYC* (*Zhan et al., 2008*) or the Hippo pathway (*Calses et al., 2019*) on overall tissue integrity and promotion of tumor growth in a none tissue wide expression setting. Clearly, to better interrogate the effect of oncogenic drivers on epithelial polarity in more detail, to understand deficiencies in cell-cell interactions towards tumorigenesis and to settle conflicting reports as mentioned above, there is a need for a more detailed analysis, employing a model system that does not show modification of all cells in the tissue.

The interaction of tumor cells with the immediate microenvironment has been subject of extensive studies with regards to immune cells (*Binnewies et al., 2018*) and other tumor associated cell types (*Tabassum and Polyak, 2015*); however, the interaction with the normal neighboring cells has not been explored in real time using an organotypic mammalian model system. Rather, questions of cell competition in heterogenous tissues have mainly been addressed in either 2D culture systems or in the *Drosophila* wing (*Tamori and Deng, 2011*). Alternatively, organ-specific mammalian cell types that self-organize in a manner similar to in vivo can now be studied in specialized 3D culture conditions in vitro (*Lancaster and Knoblich, 2014*). Furthermore, these cultures can be used to model disease and serve as an alternative system for drug testing that better recapitulate effects as compared to conventional 2D cell culture (*Clevers, 2016*; *Simian and Bissell, 2017*).

Our adaptation of such culture systems, to mark and at the same time enable single cells to express regulatable oncogenes, gave rise to the first stochastic breast tumor model using mammary luminal epithelial acini. Together with a newly established imaging and data processing pipeline, we have developed an integrated approach that allows us to follow cell fates and to interrogate cell-cell interactions of a tumor cell with the normal epithelium. We report that cells within a polarized, differentiated epithelial cell layer are able to establish tumorigenic outgrowth upon stochastic activation of strong oncogenic drivers, supporting the concept of interconversion between differentiated- and stem-like states (*Chaffer and Weinberg, 2015*; *Gupta et al., 2011*; *Plaks et al., 2015*) for tumor initiating cells. However, not all cells expressing the very powerful combination of *MYC* and *Neu* oncogenes can expand, demonstrating an inhibitory effect that needs to stem from the normal epithelial microenvironment in the presented system. The ability to distinguish marked tumor cells from the normal epithelium will now allow us to perform single-cell RNA sequencing analysis on select sorted cells. This will help delineate the signaling networks within the immediate tumor microenvironment and further clarify the molecular basis of these distinct processes.

The amenability of 3D multicellular systems to interference with small molecule inhibitors, viral shRNA vectors and genomic editing has the potential to further our understanding of the mechanisms important during tumor initiation. In the future, this will also permit drug screens on the heterogeneous populations of normal and malignant cells, as presented in the described stochastic breast tumor model. The ability to follow tumor regression and establishment of minimal residual disease within a tissue like acinus structure will help distinguish between overall cytotoxic drugs and tumor cell specific inhibitory drugs.

Taken together, we strongly believe that our integration of a true stochastic tumor model with the ability to image single-cell fates will successfully bridge the gap between genetically modified model systems and the clinical situation, helping gain novel insights on breast cancer.

# Materials and methods

## Key resources table

| Reagent type (species) or resource | Designation | Source or reference | Identifiers | Additional information |
|---|---|---|---|---|
| Strain, strain background (*M. musculus*, FVB/NJ strain) | TetO-MYC/TetO-Neu/MMTV-rtTA | *Moody et al., 2002* *Havas et al., 2017* | Tri-transgenic (T) strain | |
| Strain, strain background (*M. musculus*, FVB/NJ strain) | TetO-MYC/ TetO-Neu | *Moody et al., 2002* *Havas et al., 2017* | Bi-transgenic (B) strain | |
| Strain, strain background (*M. musculus*, FVB/NJ strain) | R26-H2B-mCherry | *Abe and Fujimori, 2013* | RIKEN, CDB0239K | |
| Recombinant DNA reagent | pWPXL plasmid | Didier Trono; Addgene | RRID:Addgene_12257 | 2nd generation lentiviral transfer plasmid |
| Recombinant DNA reagent | pLVPT-GDNF-rtTR-KRAB-2SM2 | Patrick Aebischer and Didier Trono; Addgene | RRID:Addgene_11647 | Template for pGK promoter |
| Recombinant DNA reagent | pCMV-VSV-G | Bob Weinberg; Addgene | RRID:Addgene_8454 | Envelope protein for producing lentiviral and MuLV retroviral particles |
| Recombinant DNA reagent | psPAX2 | Didier Trono; Addgene | RRID:Addgene_12260 | 2nd generation lentiviral packaging plasmid |
| Tranfected construct (*M. musculus*) | pLv-pGK-H2B-GFP | This paper | Reporter Virus | Drives expression of Reporter-GFP |
| Transfected construct (*M. musculus*) | pLv-pGK-rtTA-p2A-H2B-GFP | This paper | Inducer-reporter Virus | Drives expression of rtTA and Reporter-GFP |
| Antibody | Anti-c-MYC (D84C12) (Rabbit monoclonal) | Cell signalling Technologies | Cat. # 5605; RRID:AB_1903938 | IF dilution (1:900) |
| Antibody | Anti-alpha6-integrin (clone NKI-GoH3) (mouse monoclonal) | Millipore | Cat. # MAB1378; RRID:AB_2128317 | IF dilution (1:80) |
| Antibody | Anti-ZO1 (61–7300) (rabbit polyclonal) | Life Technologies | Cat. # 61–7300; RRID:AB_2533938 | IF dilution (1:500) |

*Continued on next page*

*Continued*

| Reagent type (species) or resource | Designation | Source or reference | Identifiers | Additional information |
|---|---|---|---|---|
| Sequence-based reagent | *Neu* Fw | This paper | PCR primers | CGTTTTGTGGTCATCCAGAACG |
| Sequence-based reagent | *Neu* Rv | This paper | PCR primers | CTTCAGCGTCTACCAGGTCACC |
| Sequence-based reagent | *MYC* Fw | This paper | PCR primers | GCGACTCTGAGGAGGAACAAGA |
| Sequence-based reagent | *MYC* Rv | This paper | PCR primers | CCAGCAGAAGGTGATCCAGACT |
| Sequence-based reagent | mCherry Fw | This paper | PCR primers | GAGGCTGAAGCTGAAGGAC |
| Sequence-based reagent | mCherry Rv | This paper | PCR primers | GATGGTGTAGTCCTCGTTGTG |
| Sequence-based reagent | *PUM11* Fw | This paper | PCR primers | AATGTGTGGCCGGATCTTGT |
| Sequence-based reagent | *PUM11* Rv | This paper | PCR primers | CCCACAGTGCCTTATACACCA |
| Commercial assay or kit | RNA PureLink Mini Kit | Thermo Fisher | Cat. # 12183018A | RNA Extraction Kit |
| Commercial assay or kit | SuperScript VILO cDNA Synthesis Kit | Thermo Fisher | Cat. # 11754050 | cDNA Synthesis Kit |
| Software, algorithm | Big Data Processor (plugin) | Fiji; *Tischer et al., 2019a* | RRID:SCR_018484 | |
| Software, algorithm | CATS (plugin) | Fiji; *Tischer and Pepperkok, 2019b* | RRID:SCR_018486 | |
| Software, algorithm | Imaris | Imaris x64 | Bitplane; RRID:SCR_007370 | Software available at: http://bitplane.com |
| Other | DAPI stain | Thermo Fisher | Cat# 62248; RRID:AB_2307445 | (1 mg/mL), (1:1000 dilution) |

## Animals

The mouse strains TetO-MYC/MMTV rtTA (*D'Cruz et al., 2001*) and TetO-Neu/MMTV rtTA (*Moody et al., 2002*), that have been previously described, were bred in order to establish the tri-transgenic strain TetO-MYC/TetO-Neu/MMTV rtTA (T) or bi-transgenic strain TetO-MYC/TetO-Neu (B). Reporter H2B-mCherry was crossed into the B and T lines using a R26-H2B-mCherry line (*Abe and Fujimori, 2013*) (RIKEN, CDB0239K). All ten mammary glands were harvested (from virgin female mice between 8–10 weeks old), digested and singularized for establishing acinar cultures. All mice used in this study were housed according to the guidelines of the Federation of European Laboratory Animal Science Associations (FELASA).

Rational for the use of these oncogenes: *ERBB2* is overexpressed in ~20% of breast cancers (*Arteaga et al., 2012*), *MYC* in 15–50% of human breast cancer (*Blancato et al., 2004*). The combination of *MYC* and *ERBB2* is found in highly aggressive human breast cancer (*Al-Kuraya et al., 2004*):and - in fact- *Neu* (the rodent homolog for the human *ERBB2* gene used in this mouse model) and *MYC* strongly accelerate tumor onset In the combined transgenic animals (average 45 days) as

compared to single transgenic animals (*MYC* 155 days, *Neu* 99 days), In all cases tumors regress rapidly to non-palpable state following oncogene silencing.

## Lentivirus cloning and production

The lentivirus design is based on pWPXL backbone, which was a gift from Didier Trono (Addgene #12257). The coding region from the original plasmid was excised using ClaI and NdeI in order to insert a new multiple cloning site (MCS). The pGK promoter was PCR amplified from pLVPT-GDNF-rtTR-KRAB-2SM2, which was a gift from Patrick Aebischer and Didier Trono (Addgene #11647) and cloned using XhoI and EcoRI restriction sites. For the plasmid pLenti-rtTA-GFP the synthetic region rtTA-p2A-H2B-GFP was cloned downstream of the pGK promoter using EcoRI and NheI sites. The plasmid pLenti-Null-GFP is derived from the pLenti-rtTA-GFP by removing the rtTA sequence, using the restriction sites EcoRI and BamHI, and retaining H2B-GFP in the coding region. For production of lentivirus particles, we seeded $1.6 \times 10^7$ HEK-293T cells (Lenti-X - Clontech Cat. # 632180) in 500 $cm^2$ square dishes (Corning Cat. # 431110). After 24 hr, the cells were supplemented with medium containing 25 µM of chloroquine diphosphate (Sigma-Aldrich Cat. # C6628). After a 5 hr incubation, using 360 µg of polyethyleneimine (4 µg for each µg of plasmid), we transfect the cells with a mixture of endotoxin free plasmids: 20 µg pCMV-VSV-G (Addgene #8454); 30 µg psPAX2 (Addgene #12260); 40 µg transfer plasmids pLenti-rtTA-GFP or pLenti-Null-GFP. We harvested the medium after 48 hr, 72 hr and 96 hr after transfection. Concentration of the lentivirus from the collected medium was performed using an ultracentrifuge (Beckman Sw32 rotor) at 25,000 rpm for 2 hr at 4°C. The lentivirus pellet was resuspended in 1000 µl of HBBS buffer, aliquoted and stored at −80°C. The lentivirus titer was measured using FACS analysis as described by Kutner and colleagues (*Kutner et al., 2009*).

## 3D acinus cultures

Mammary glands harvested from mice (see above), were digested in order to prepare a single-cell solution. For this, the tissue was divided in four loosely capped 50 ml falcon, each supplemented with 5 ml serum-free media (DMEM/F12 supplemented with 25 mM HEPES and 1% Pen Strep(100 U/ml Penicillin; 100 µg/ml Streptomycin; ThermoFisher Cat. # 15140122)) and 750 U of Collagenase Type 3 (Worthington Biochemical Corp Cat. # LS004183), 20 µg of Liberase (Roche Cat. # 5401020001) and incubated overnight at 37°C and 5%$CO_2$. The glands were then mechanically disrupted using a 5 ml pipette, and washed in PBS before being pelleted at 1000 rpm for 5 min. The cell pellet was resuspended in 5 ml of 0.25% Trypsin-EDTA and incubated for 45 min at 37°C and 5% $CO_2$. The enzymatic reaction was then neutralized using 40 ml of serum supplemented medium (DMEM/F12 with 25 mM HEPES, 1% Pen Strep and 10% FBS Tetracycline Free certified (Biowest Cat. # S181T). The cells were pelleted again, resuspended in Mammary Epithelial Cell Basal Medium (PromoCell Cat. # C-21210) and seeded in collagen coated plates (Corning Cat. # 354400) overnight at 37°C and 5%$CO_2$. This allows for epithelial cells to adhere to the surface of the plates while the other cell types float on top in the medium and can be easily removed by vacuum suction. The epithelial cells were detached from the collagen coated plates by incubating them with 0.25% Trypsin-EDTA for 5–7 min at 37°C and 5%$CO_2$, following inactivation with serum supplemented media. The single-cell solution was pelleted, resuspended in MEBM and counted. We mixed 50,000 cells with 90 µl of Matrigel Matrix basement Membrane growth factor reduced phenol red free (Corning Cat. # 356231), and seeded this mixture into a 12 well plate (Corning Cat. # 3336) and incubated it for 30–40 min until the matrigel solidified. The gels were supplemented with 1.5 ml MEBM and allowed to grow at 37°C and 5%$CO_2$.

For transduction, after 3 days of growth, the gels were mechanically disrupted and placed in a 15 ml falcon. Two disrupted gels were placed in one 15 ml falcon with 2 ml of MEBM supplemented with 25U of Collagenase type I and 5 µg of Liberase. Following incubation in this solution for 2 hr at 37°C and 5%$CO_2$, when the matrigel was totally digested, the acini were washed 3 times with 15 ml of serum supplemented media and once with 15 ml of serum free media, and pelleted at 1000 rpm for 5 min. We then supplemented the acinus pellet (from two original gels) in 10 µl of MEBM and added $6 \times 10^5$ lentivirus particles to the solution. We then mixed this solution with 90 µl matrigel and plated it in 35 mm dishes (Greiner Bio-One Cat. # 627160) and placed in incubator for 30–40

min until the matrigel solidified. The gels were supplemented with 3 ml MEBM and incubated for 2 days at 37°C and 5%CO$_2$ in order to allow for acinus recovery and lentiviral gene expression.

For induction of oncogenes in the cells of the acini, doxycycline (Sigma Cat. # D9891) was supplemented in the media. 800 ng/ml of doxycycline was used to induce T acini and 600 ng/ml was used for B acini. qPCR analysis was used to standardize the doxycycline dosage for B acini (see below).

## qPCR analysis

The qPCR technique was performed following the MIQE guidelines, where the total RNA was isolated from the mammary gland acini using RNA PureLink Mini Kit (ThermoFisher Cat. # 12183018A) and 2.5 ug was reverse transcribed to cDNA using SuperScript VILO cDNA Synthesis Kit (ThermoFisher Cat. # 11754050). Using Primer3 software we designed specific primers for DNA intercalating fluorescent dye approach for the transgenes *Neu* (Forward: CGTTTTGTGGTCATCCAGAACG and Reverse: CTTCAGCGTCTACCAGGTCACC) and c- *MYC* (Forward: GCGACTCTGAGGAGGAACAAGA and Reverse: CCAGCAGAAGGTGATCCAGACT). As endogenous controls, mCherry (Forward: GAGGCTGAAGCTGAAGGAC and Reverse: GATGGTGTAGTCCTCGTTGTG) and Pum1 (Forward: AATGTGTGGCCGGATCTTGT and Reverse: CCCACAGTGCCTTATACACCA) were used. Primer efficiency was verified and established between 95% and 105% Each sample was analyzed in duplicate and non-template controls were used in each qPCR run. Analyses were carried out using a StepOne device (Applied Biosystems, USA). Analysis of relative gene expression data was performed according to the $2−ΔΔCq$ method and the results were expressed as fold change of $ΔΔCq$ values obtained from the reference T800 acini (*Figure 1—figure supplement 1*).

## Immunofluorescence staining

Matrigel cultures were grown as described above and plated on Nunc Lab-Tek II (Thermo Cat. # 155382) chambers. At pre-defined timepoints, the gels were fixed using 4% PFA for 2–3 min, following 3 washes with PBS. The gels were blocked with 10% goat serum for 2 hr at room temperature, followed by incubation with primary antibodies was done overnight at 4°C. The remaining immunofluorescence staining was performed as per standard protocol for c-MYC (Cell Signaling Technologies, Cat. # D84C12, 1:900), alpha6-integrin (Millipore Cat. # MAB1378, dilution 1:80) and ZO1 (Life Technologies Cat. # 61–7300, dilution 1:500). The nuclei were counter stained with 1:1000 DAPI (ThermoFisher Cat. # 62248, 1 mg/ml, dilution 1:1000) and mounted in anti-fading mounting medium (VECTASHIELD Mounting Medium with DAPI (Vecto Cat. # H1500-10)). Please note that the c-MYC antibody (Cell Signaling Technologies, Cat. # D84C12) recognizes specifically the human protein, which is transgenically expressed and does not recognize endogenous mouse MYC protein.

Stained gels were imaged on Leica SP5 confocal microscope using 63x water lens and the LAS AF imaging software.

## Light sheet microscopy

### Sample holder preparation and mounting

Imaging was performed on the InVi SPIM inverted light-sheet microscope (Luxendo Light-Sheet, Bruker Corporation). Sample mounting for the InVi SPIM is suitable for 3D matrigel cultures that are used to grow and transduce mammary acini (see above). The sample holder is made of medical grade plastic (PEEK). A 25 µm thin membrane (FEP; Luxendo) with a refractive index matching that of water is glued to the upper surface of a groove in the sample holder with a biocompatible silicone glue (Silpuran 4200; Wacker), forming a trough with transparent bottom (*Figure 3—figure supplement 1*). Matrigel cultures were carefully cut with a scalpel into rectangular slivers and transferred onto the FEP membrane's trough. Once the gel sliver was aligned in place, 20–30 µl of fresh matrigel drops were poured onto the gel sliver in the sample holder until there was a thin layer of liquid matrigel on top of the gel sliver. The setup was incubated for 20 min at 37°C in a 5% CO$_2$ incubator to allow the matrigel layer on top to solidify. Once the gel was solidified, 600–800 µl of MEBM supplemented with/without doxycycline was added to the sample holder's FEP sheet trough. Preferably, freshly mounted sample gels were allowed to settle overnight in the incubator to prevent any gel drift during imaging, when the holder is placed into the imaging chamber of the microscope. The imaging chamber acts as an incubator with environmental control and it has a reservoir for

immersion medium, which is filled with water so that both objective lenses and the bottom of the sample holder are below the water surface (*Figure 3—figure supplement 1*).

## Imaging configuration and conditions

The InVi SPIM is equipped with a Nikon CFI 10x/0.3NA water immersion lens for illumination and a Nikon CFI-75 25x/1.1NA water immersion lens for detection. For excitation of GFP and mCherry, 488 nm and 594 nm laser lines were used, respectively, while emission was selected using a 497–554 nm band pass filter and a 610 nm long pass filter, respectively. 3D image stacks were acquired with a light-sheet thickness of 4 µm, a final magnification of 62.5x, resulting in 104 nm pixel size. The In-Vi SPIM environmental control was set to 37°C, 5% $CO_2$ and 95% humidity. A series of optimization experiments, involving different laser powers, exposure times and z-step sizes yielded laser powers of 13 µW for 488 nm and 36 µW for 594 nm, 100 millisecond exposure time per frame and 1 µm z-spacing between frames to be optimal for long-term imaging (96–120 hr) without photo-bleaching or photo-toxic effects on growth.

Images were recorded as 2D planes ranging from 100 to 500 in number, depending on the acinus size. Each 3D stack of planes was recorded in 2 channels - mCherry (all cells) and GFP (transduced cells). Depending on the duration of the time lapse imaging, 450–600 image stacks (equivalent to ~72–96 hr) were recorded per acinus at 10 min intervals.

## Image analysis

Big Data Processor (*Tischer et al., 2019a*), a Fiji plugin for lazy loading of big image data, was used to visualize the images in 2D slicing mode, crop stacks in x, y, z, and t, bin images (3 × 3×1 in x, y, z), perform chromatic shift correction between channels and convert. h5 files from the InVi SPIM into an Imaris compatible multi-resolution file format (.ims) for further analysis (*Figure 3—figure supplement 3*).

The oncogenic cells (H2B-GFP channel) displayed heterogeneous morphologies as well as varying intensity textures, making it difficult to segment them using conventional thresholding approaches. We thus used a trainable segmentation approach to convert the raw intensity values into pixel probability maps, using the Fiji plugin CATS (*Tischer and Pepperkok, 2019b*) (Context Aware Trainable Segmentation). Using the H2B-GFP channel images as input, we trained three pixel classes: background, nucleus center and nucleus boundary. For training we drew about 20(background), 120 (nucleus center), 100(nucleus boundary) labels distributed across the different time-frames of the movie. After feature computation and training of a Random Forest classifier the whole dataset was processed on EMBL's high performance computer cluster. The segmentation of one dataset -typically 100 timepoints- is distributed across few hundred jobs, each job using 32 GB RAM, 16 cores, and running for about 30 min. The nucleus center probability maps were then exported from CATS and added as an additional channel to the converted intensity data (*Figure 3—figure supplement 3*).

The data were then loaded into Imaris (*Bitplane AG, 2020*, Software available at http://bitplane.com) for 3D visualization and further processing. Using the Imaris' Surfaces function, we segmented the nucleus center probability maps into objects. To do so, probability maps were manually thresholded, using a surface smoothening parameter of 0.3 µm; the minimum quality parameter for seed points was set to 0.1, and object splitting was applied for objects larger than 5.5 µm. Objects with volumes less than 20 $µm^3$ were excluded. Next, all objects were tracked over time using Imaris' Lineage tracking algorithm with a maximum distance between objects in subsequent time points limited to 10 µm and a maximum gap size between identification of the object in a particular track limited to 10 time points. Analysis of segmentation results is shown in *Figure 4—figure supplement 1*. Most of the errors in the object segmentation were false merges, where two cells were segmented as one. This kind of error is frequently not sustained in the previous or following time points and the maximum gap size parameter of the tracking algorithm thus frequently provides correct tracks nonetheless. The resulting lineage trees of proliferating tumor cells within the acinus were corrected manually within Imaris, for example, excluding apoptotic cells and auto-fluorescent debris.

Center of mass coordinates of each cell were measured and exported from Imaris for subsequent feature analysis (*Figure 4b*).

## Feature analysis

Observations suggest that tumors in acini originate from clusters of oncogene-expressing cells produced by independent transduction events. To identify these clusters, we computed the pairwise Euclidean distances between all oncogene-expressing cells in an acinus at the start of the experiment and applied hierarchical clustering with complete linkage. Clusters were identified automatically by cutting the branches of the trees using the dynamic tree cut algorithm (*Langfelder et al., 2008*). This defined a cluster as a group of oncogene-expressing cells that are closer to each other than to other oncogene-expressing cells of the same acinus. Note that a cluster can be composed of a single cell if this cell is comparatively isolated from other transduced cells. For each cluster we identified the following features as possibly linked to tumor formation: (1) number of cells in the acinus (2) cell density expressed as the ratio of number of cells to acinus surface area computed by assuming the acinus is a sphere with diameter equal to the distance between the two most distant cells (3) number of oncogene-expressing cells in the acinus (4) number of cells (including both oncogene-expressing and normal cells) in the cluster volume defined as the sphere centered at the center of mass of the cluster with diameter equal to the distance between the two farthest oncogene-expressing cells of the cluster (5) number of oncogene-expressing cells in the cluster (6) average pairwise distance between all cells in the cluster volume (7) average pairwise distance between oncogene-expressing cells in the cluster (8) fraction of oncogene-expressing cells in the cluster volume (9) number of contacts between oncogene-expressing cells in the cluster. Two cells are presumed in contact if they are less than the average cell diameter + 2 standard deviation apart.

Oncogene-expressing cells were tracked over time and a cluster was associated with a tumor outcome if any of its cells lead to tumor formation. Here, 'tumor formation' is defined as the phenotypic observation of increased proliferation rate and disrupted polarity observed in transduced cell clusters upon induction with doxycycline. Regions displaying 'tumor formation' within the acinus often form multi-layer cell clusters with fast-dividing cells and increased apoptosis. To identify which features were linked to this outcome, we took an information-theoretic approach to model selection. We fitted a logistic regression model for all possible linear combinations of features and selected the best model based on the Akaike information criterion (with correction for small sample sizes) (*Calcagno and Mazancourt, 2010*). This model included the following features: number of oncogene-expressing cells in the cluster, number of oncogene-expressing cells in the acinus, number of cells in the cluster and number of contacts between oncogene-expressing cells in the cluster. Of these features, only the number of oncogene-expressing cells in the cluster contributed significantly to tumor formation with an odds ratio of 8.96 (*Figure 4b*). Of note, all the models within 3 information criterion units of the best model also included the number of oncogene-expressing cells in the cluster as a significant contributor to tumorigenesis. Relative importance of different features can be estimated by the sum of the information criterion values of the models in which each feature appears (*Buckland et al., 1997*). Computing relative variable importance across all models also indicated this feature as the most important. To control for acinus of origin, we also evaluated a mixed effect model including the three most important fixed effect features (number of transduced cells in a cluster, number of cells in a cluster and number of contacts between transduced cells in a cluster), a random effect for acinus and interaction terms for the starting number of normal cells. In this case again, the number of oncogene-expressing cells in the cluster appeared as the main contributing factor to tumorigenesis with an odds ratio of 6.73 (*Figure 4—figure supplement 2b*).

## Acknowledgements

The authors want to thank Sylwia Gawrzak, Ksenija Radic, Rocio Sotillo, Robert Prevedel and Jan Ellenberg for critically reading the manuscript and Marta Garcia Montero for mouse husbandry. This study was technically supported by EMBL Advanced Light Microscopy Facility (ALMF) and the EMBL Laboratory for Animal Resources (LAR).

# Additional information

## Competing interests

Monika Loeschinger: Monika Loeschinger is employed by Luxendo GmbH, FM BU, Bruker Nano Surfaces, Heidelberg, Germany, the manufacturer of the InVi SPIM light-sheet microscope. Malte Wachsmuth: Malte Wachsmuth is employed by Luxendo GmbH, FM BU, Bruker Nano Surfaces, Heidelberg, Germany, the manufacturer of the InVi SPIM light-sheet microscope. The other authors declare that no competing interests exist.

## Funding

No external funding was received for this work.

## Author contributions

Ashna Alladin, Data curation, Formal analysis, Investigation, Visualization, Writing - original draft, Writing - review and editing, Implemented and performed the SPIM imaging and implemented the image analysis workflows, Performed IF analysis; Lucas Chaible, Conceptualization, Data curation, Performed the cloning; Lucia Garcia del Valle, Data curation, Formal analysis, Performed SPIM imaging and analyzed data; Reither Sabine, Monika Loeschinger, Data curation, Validation; Malte Wachsmuth, Supervision; Jean-Karim Hériché, Software, Formal analysis, Writing - review and editing; Christian Tischer, Resources, Software, Formal analysis, Supervision, Visualization, Writing - review and editing; Martin Jechlinger, Conceptualization, Supervision, Validation, Writing - original draft, Project administration, Writing - review and editing

## Author ORCIDs

Jean-Karim Hériché ⓘD http://orcid.org/0000-0001-6867-9425
Martin Jechlinger ⓘD https://orcid.org/0000-0002-3710-4466

## Ethics

Animal experimentation: Animals are treated at the European Molecular Laboratory in agreement with National and International laws and policies. All effort are made to use the minimal amount of animals as possible in accordance with Russell and Burch's (1959) principle of (3Rs) reduction and highest ethical standards. The IACUC (Institutional Animal Care and Use Committee) approved the work with these mice (approval # MJ160070).

## Decision letter and Author response

Decision letter https://doi.org/10.7554/eLife.54066.sa1
Author response https://doi.org/10.7554/eLife.54066.sa2

# Additional files

## Supplementary files

• Source code 1. Transduced Cell Cluster - Feature Analysis.

• Supplementary file 1. Comprises of an html file describing the *Source code 1* in 'Feature_Analysis. Rmd'.

• Supplementary file 2. Comprises of an .xlsx file with 20 sheets, one for each acinus analyzed, and contains the x,y,z coordinates for each cell in the respective acinus at the beginning of the SPIM recording. This file was input into the source code to carry out the acinus feature analysis described in *Figure 4b* and *Figure 4—figure supplement 1*.

• Supplementary file 3. Comprises of an .xlsx file with 20 sheets, one for each acinus analyzed, and contains the x,y,z coordinates for each cell in the respective acinus at the end of the SPIM recording. This file was input into the source code to carry out the acinus feature analysis described in *Figure 4b* and *Figure 4—figure supplement 1*.

- Supplementary file 4. Comprises of an .xlsx file with 20 sheets, one for each acinus analyzed, and contains the 'label' for each transduced cell (corresponding to the labels in *Supplementary file 2*) in the respective acinus at the beginning of the SPIM recording. This file was input into the source code to carry out the acinus feature analysis described in *Figure 4b* and *Figure 4—figure supplement 1*.

- Transparent reporting form

### Data availability

1) Entire image recordings (movies) of time-lapse panels in Figure 3a and 3b (3 Video files in total) have been provided as supplementary movie files. 2) We have uploaded the code for the Feature analysis of the nine acinar features described in Figure 4, as source code file "Feature_Analysis. Rmd". Refer to Supplement file 1 and Online Materials and Methods section for analysis summary. 3) We have uploaded the html file describing the source code as Supplementary file 1. 4) Three. xlsx files with 20 sheets each, one sheet for each acinus analyzed are provided as Supplementary files 2, 3, and 4. These contain the x,y,z coordinates for each cell in the respective acinus at the beginning of the SPIM recording (Supplementary file 2) and at the end (Supplementary file 3). Supplementary File 4 contains the "label" for each transduced cell (corresponding to the labels in Supplement File 2) for the acini at the beginning of the SPIM recording. These. xlsx files were input into the source code to carry out the acinus feature analysis described in Figure 4b and Figure 4 - figure supplement 1. 4) We have deposited the original imaging data for all acini recorded and analyzed (20 mammary acini) at the BioStudies archive at EMBL-EBI (https://www.ebi.ac.uk/biostudies/studies/S-BIAD13). A total of 390-450. h5 image files recorder from 2 channels on the microscope are uploaded for each acini (10 minute time intervals). Raw image data from the microscope was cropped to remove empty pixels, binned in x,y (3,3) and converted to 8-bit images using Big Data Processor Fiji Plug in (http://doi.org/10.5281/zenodo.2574702). This data repository also contains video files generated via Imaris for each acinus, showing fluorescence SPIM miscropscopy data (pre-processed raw files available in respective folders) in 2-color 3D projections (mcherry- magenta; GFP- green) for observing visual phenotypes.

The following datasets were generated:

| Author(s) | Year | Dataset title | Dataset URL | Database and Identifier |
|---|---|---|---|---|
| Tischer C, Norlin N, Pepperkok R | 2019 | BigDataProcessor: Fiji plugin for big image data inspection and processing | http://doi.org/10.5281/zenodo.2574702 | Zenodo, 10.5281/zenodo.2574702 |
| Alladin A, Chaible L, Garcia del Valle L, Sabine R, Loeschinger M, Wachsmuth M, Hériché J-K, Tischer C, Jechlinger M | 2020 | Tracking the cells of tumor origin in breast organoids by light sheet microscopy - SPIM movie data | https://www.ebi.ac.uk/biostudies/studies/S-BIAD13 | BioStudies, S-BIAD13 |

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
