## [Decision Letter]

**Acceptance summary:**

This paper demonstrates a beautiful methodology for tracking tumour formation within 3D acinar structures, with single cell resolution. It also suggests that tumour formation may depend upon local interactions between transformed cells. The method described may be used to better understand the underpinnings of stochastic tumor evolution.

**Decision letter after peer review:**

Thank you for submitting your article "Tracking cells in a novel organoid model by light sheet microscopy reveals proximity effects in breast cancer initiation" for consideration by *eLife*. Your article has been reviewed by three peer reviewers, one of whom is a member of our Board of Reviewing Editors, and the evaluation has been overseen by Richard White as the Senior Editor. The following individual involved in review of your submission has agreed to reveal their identity: Anne Rios (Reviewer #3).

The reviewers have discussed the reviews with one another and the Reviewing Editor has drafted this decision to help you prepare a revised submission.

Summary:

This manuscript by Jechlinger and colleagues combines a novel transgenic organoid model of breast hyperplasia with light-sheet microscopy in order to demonstrate that hyperplastic outgrowths occur most readily when transformed cells are adjacent to each other. Specifically, the authors compared early progression (hyperplastic growth and depolarization) in organoids wherein all versus only a subset of epithelial cells harbor the *Neu* and MYC driver mutations. Aided by time lapsed light-sheet microscopy and computational analyses, they surmised that the number of transduced cells in a cluster dictates progression, suggesting that special cues are likely required for cancer progression to occur. This work was reviewed by 2 experts and myself. One of the reviewers was notably an expert in the imaging and characterization of organoids, and thus able to comment on the technical aspects of this study, which given its focus, are the most important.

Overall, the reviewers felt that the paper describes an interesting tool that is accompanied by beautiful images and a fairly compelling analysis. However, in order to better justify the biological conclusions, additional experiments would be needed. These would include an analysis of cells which outgrow (versus those that do not). Such studies may include single cell RNA sequencing to determine how cells in clusters differ from those which are more isolated, as well as a more thorough assessment of polarity and consideration of microenvironmental cues such the concentration of myo-epithelial signals. Hence, the main issues raised by the reviewers related to the claims that a distinct number of initiator cells can explain tumorigenic clone expansion and that a proximity-controlled interaction or signaling network between different transformed cells is required for tumor outgrowth in a normal epithelium. In order to substantiate these claims, a number of experiments would have to be done. I would suggest, rather, that the claims are tempered and that the suggested technical revisions are completed.

Essential revisions:

1) Heterogeneities associated with the growth of organoids in general should be considered with additional controls: The authors cannot claim that you need interaction/cooperation between different transformed cells when the “organoid” with few activated cells clearly show an overall growth slower or even non-existent compare to the one with expansion of tumorigenic clones (organoid size, cell compactness). Differences of growth and growth advantage for some “organoids” while culturing is quite common. The authors should think of a comparison of growth dynamics of untransduced tissue and provide the initial volume in both conditions as well as assess the impact of proliferation differences between different organoids.

2) The idea that the initial cluster size dictates tumor formation should be better supported: If indeed tumor formation is dependent on the initial cluster size, this effect should also be detectable at single organoid level: A single organoid should have clusters formed by several cells that will give rise to a tumor and clusters formed by only 1 cell that will not give rise to a tumor. The authors should define such examples, and fit a model with random effect, where each organoid will be considered as random effect. This could as well exclude the possibility that the results are due to the fact that organoids have a differential growth capacity.

3) The AI approach should be strengthened and/or better articulated: In the logistic regression model used to assess which feature is responsible for the tumor formation, authors should include interaction effects between different features, which is not considered by the linear model used. As indeed the size of the organoids could influence the number of cells per cluster, etc.

4) The conclusions must be either softened or expanded upon (with experiments such as those outlined in the summary): Considering that the conclusion of the authors would be correct and that the tumor formation is solely dependent on the cluster size, please provide an explanation why is it independent of the interactions between cells of the same cluster. It would be logical to assume that cells have to be in close contact to exert an effect on each other.

5) The semantics regarding "organoids" should be addressed: The authors call “organoids”, the in vitro 3D culture they use. Not all 3D spheroids culture can be called “organoids”, it is essential first to examine if the 3D structures grown in culture resemble the original breast tissue (e.g. cell composition, geometry). Why not use the well-defined protocol developed for mouse organoid culturing where both main mammary epithelial cell types can be modelled (luminal and myoepithelial) (Jamieson, Dekkers et al., 2016)? The authors should either prove that their cultures can be called organoids or use the right protocol. It is of great importance as they are claiming they use a stochastic model. Looking at shape and polarity, I presume their organoids are luminal-enriched.

6) Studies regarding polarity should be strengthened: 3D z-stack and rendering should have been performed to make conclusive interpretation in Figure 2A about disrupted polarity and random lumen. 2D often lead to a wrong interpretation of architectural phenotypes. Moreover, the authors should expand the polarity analyses by measuring a number of polarity markers (ZO-1, Par3).

7) The authors should refine terminology related to localized transformation and/or perform the experiments that would allow them to suggest that transformation is in fact localized: The authors also claimed a model of localized transformation while what they did is random and low rate transformation using a doxycycline inducible model with lentiviral delivery of rtTA. It is a confusing claim as localized transformation implies you control the location. This can be performed with optogenetic technology. Alternatively, the terminology used should be refined.

8) Please ensure the supplementary figures are uploaded.

9) The authors should not suggest that a big-data analysis was conducted and/or should apply more robust analytical techniques: Authors claim that they do big-data analysis, however only 20 organoids are segmented and tracked. The authors should either provide a more robust analysis or change the semantics that are used in the paper.

Reviewer #1:

This manuscript by Jechlinger and colleagues combines a novel transgenic organoid model of breast hyperplasia with light-sheet microscopy in order to demonstrate that hyperplastic outgrowths occur most readily when transformed cells are adjacent to each other. Specifically, the authors compared early progression (hyperplastic growth and depolarization) in organoids wherein all versus only a subset of epithelial cells harbor the *Neu* and MYC driver mutations. Aided by time lapsed light-sheet microscopy and computational analyses, they determined that the number of transduced cells in a cluster dictates progression, suggesting that special cues are likely required for cancer progression to occur. Overall, this is an interesting tool, accompanied by beautiful images and a fairly compelling analysis. However, in order to solidify what is still mostly a descriptive finding, an analysis of cells which outgrow (versus those that do not) would be needed.

As indicated by the authors, single cell RNA sequencing should be done to determine how cells in clusters differ from those which are more isolated. Alternatively, key factors may be analyzed perhaps with IHC, to determine if there are tangible differences.

Reviewer #2:

This is an interesting paper using real time imaging and a retroviral based strategy to induce both the MYC/activated *Neu* in single cell. The core observation is that clusters of oncogenes expressing mammary epithelial cells go on to form tumors whereas single cell isolates do not. The authors conclude that adjacent normal cells (untransduced cells) negatively inhibit tumor outgrowth. In general, the data presented support the author's conclusions. However, the authors should address the following issues before publication.

1) Although the data indicate that single cell expressing both oncogenes have normal polarity, using a number of polarity markers (ZO-1, Par3) can the authors exclude subtle alterations in polarity program.

2) What impact would co-cultivation of myoepithelial cells have on the observed tumor dynamics.

Reviewer #3:

Alladin et al. proposed a stochastic breast tumor doxycycline inducible system using mouse breast 3D cultures to establish lower cell transformation rate. They performed lightsheet time-lapse imaging together with tracking analysis to examine clone expansion overtime after transformation. The authors claimed that distinct number of initiator cells can explain tumorigenic clone expansion. In addition, they suggest that a proximity-controlled interaction or signaling network between different transformed cells is required for tumor outgrowth in a normal epithelium. I have serious concerns about this claim and even if the claim is correct, the paper does not provide any explanation for such a phenomenon. Please see my comments below, I hope it may help improve your manuscript.

– The authors cannot claim that you need interaction/cooperation between different transformed cells when the “organoid” with few activated cells clearly show an overall growth slower or even inexistent compare to the one with expansion of tumorigenic clones (organoid size, cell compactness). Differences of growth and growth advantage for some “organoids” while culturing is quite common. The authors should think of a comparison of growth dynamics of untransduced tissue and provide the initial volume in both conditions as well as assess the impact of proliferation differences between different organoids.

– If indeed tumor formation is dependent on the initial cluster size, this effect should also be detectable at single organoid level: A single organoid should have clusters formed by several cells that will give rise to a tumor and clusters formed by only 1 cell that will not give rise to a tumor. The authors should define such examples, and fit a model with random effect, where each organoid will be considered as random effect. This could as well exclude the possibility that the results are due to the fact that organoids have a differential growth capacity.

– In the logistic regression model used to assess which feature is responsible for the tumor formation, authors should include interaction effects between different features, which is not considered by the linear model used. As indeed the size of the organoids could influence the number of cells per cluster, etc.

– Considering that the conclusion of the authors would be correct and that the tumor formation is solely dependent on the cluster size, please provide an explanation why is it independent of the interactions between cells of the same cluster. It would be logical to assume that cells have to be in close contact to exert an effect on each other.

– The author call “organoids”, the in vitro 3D culture they use. Not all 3D spheroids culture can be called “organoids”, it is essential first to examine if the 3D structures grown in culture resemble the original breast tissue (e.g. cell composition, geometry). Why not use the well-defined protocol developed for mouse organoid culturing where both main mammary epithelial cell types can be modelled (luminal and myoepithelial) (Jamieson, Dekkers et al., 2016)? The authors should either prove that their cultures can be called organoids or use the right protocol. It is of great importance as they are claiming they use a stochastic model. Looking at shape and polarity, I presume their organoids are luminal-enriched.

– 3D z-stack and rendering should have been performed to make conclusive interpretation in Figure 2A about disrupted polarity and random lumen. 2D often lead to a wrong interpretation of architectural phenotypes.

– The authors also claimed a model of localized transformation while what they did is random and low rate transformation using a doxycycline inducible model with lentiviral delivery of rtTA. It is rather confusing claim as localized transformation implies you control the location. This can be performed with optogenetic technology.

– Important to mention: No supplementary figures were uploaded.

– Authors claim that they do big-data analysis, however only 20 organoids are segmented and tracked, this is misleading. The authors should either provide a more robust analysis or change the semantics that are used in the paper. Also, few terabytes per organoid imaged…I may advise to consider other imaging technologies such as confocal that will give better images for less data generated.

---

## [Author Response]

Essential revisions:1) Heterogeneities associated with the growth of organoids in general should be considered with additional controls: The authors cannot claim that you need interaction/cooperation between different transformed cells when the “organoid” with few activated cells clearly show an overall growth slower or even non-existent compare to the one with expansion of tumorigenic clones (organoid size, cell compactness). Differences of growth and growth advantage for some “organoids” while culturing is quite common. The authors should think of a comparison of growth dynamics of untransduced tissue and provide the initial volume in both conditions as well as assess the impact of proliferation differences between different organoids.

We would like to thank the reviewers for bringing up this crucial point. We have addressed this question via an in-depth analysis of single organoid features, inclusion of 2 additional supplementary figures and new text in the manuscript:

a) Analysis of numbers and spatial coordinates of all (transduced and normal) cells of each acinus at the start and the end of imaging allowed us to “analyze the effect of whole acinar features at the beginning and end of the imaging period to try and detect acinus-specific advantages, like acinus size, cell density and cell proliferation rate. Acini with transduced cell clusters that displayed a tumor outcome were compared to acini where the cell clusters did not result in tumor outcomes (Figure 4—figure supplement 1A). Acini with tumors only differed significantly from those without tumor on the number and proliferation rate of transduced cells in line with the observation that clusters of transduced cells are the primary indicators of tumor outcome.”

b) We illustrate this observation further by inclusion of an example of “similar sized acini bearing the same number of transduced cells that showed differential cluster expansion depending on the numbers of transduced cells within a cluster” (Figure 4—figure supplement 2; comparison A to B).

c) To account for acinus specific effects, we “also evaluated a logistic regression model including the three most important features identified (above) plus a random effect for acinus of origin and interaction terms for the starting number of normal cells in the acinus. In this model again, the number of oncogene-expressing cells in the cluster appeared as the main contributing factor to tumorigenesis with an odds ratio of 6.7 (Figure 4—figure supplement 1B).”

d) We include 3D projection videos over time (uploaded to https://www.ebi.ac.uk/biostudies/studies/S-BIAD13?key=d65c53a7-253d-4e4e-82df-d27be3a836f) for all 20 analyzed acini (150 transduced cells in total) to give the reader a transparent representation of the data and the possibility to examine valid concerns like this one, with ease.

2) The idea that the initial cluster size dictates tumor formation should be better supported: If indeed tumor formation is dependent on the initial cluster size, this effect should also be detectable at single organoid level: A single organoid should have clusters formed by several cells that will give rise to a tumor and clusters formed by only 1 cell that will not give rise to a tumor. The authors should define such examples, and fit a model with random effect, where each organoid will be considered as random effect. This could as well exclude the possibility that the results are due to the fact that organoids have a differential growth capacity.

We entirely agree with the reviewer’s notion that a single acinus should give rise to different phenotypes (proliferative versus non-proliferative), depending on the respective cluster/microenvironment.

a) In fact, we present one example acinus (Figure 4A) that bears expanding clusters originating from several transduced cells in vicinity (red and orange circled) as well as non-expanding clusters (blue circles; single cells or duplets within the cluster). Cell linages are shown over the imaging period.

b) We also present 2 further cases in Figure 4C (upper panels and middle panels).

c) In addition, as suggested by the reviewers, we included a more in-depth analysis of two acini in Figure 4—figure supplement 2A, showing cell lineages over time.

3) The AI approach should be strengthened and/or better articulated: In the logistic regression model used to assess which feature is responsible for the tumor formation, authors should include interaction effects between different features, which is not considered by the linear model used. As indeed the size of the organoids could influence the number of cells per cluster, etc.

We strengthened the computational modelling approach and added onto the initially presented “logistic regression model for all possible linear combinations of features”, while also tuning down the interpretation of the results:

a) “Since other models within a few information criterion units of the best one can be considered equally good at predicting tumor formation, we also looked at features included in models within 3 information criterion units from the best one. […]. It is important to note that this result does not rule out contribution from other factors which more targeted experiments may be able to reveal.”

b) We included a mixed effect model for “a random effect for acinus of origin and interaction terms for the starting number of normal cells in the acinus.” (see rebuttal point 1c). Again, here we obtained "number of transduced cells in the cluster" as the most prominent determinant.

4) The conclusions must be either softened or expanded upon (with experiments such as those outlined in the summary): Considering that the conclusion of the authors would be correct and that the tumor formation is solely dependent on the cluster size, please provide an explanation why is it independent of the interactions between cells of the same cluster. It would be logical to assume that cells have to be in close contact to exert an effect on each other.

We do not intend to claim that tumor formation is solely dependent on the number of transduced cells in a cluster, rather that this factor is a potent indicator of tumorigenic outcome when compared to the features we analyzed in Figure 4B. We have endeavored to make this clearer by inserting the following statements:

a) “However, the only feature with significant contribution to tumor formation is "number of transduced cells in the cluster". It is important to note that this result does not rule out contribution from other factors which more targeted experiments may be able to reveal.”; see also rebuttal point 3b).

b) “According to our modeling attempts the simple parameter of “number of transduced cells within a cluster” is enough to impinge on overall epithelial integrity and most likely already accounts for sub-features like “distances between transduced cells in a cluster volume” or “number of cell-cell contacts between transduced cells in a cluster volume”, which did not pan out as predictive factors on their own. […] The questions outlined above can now be assessed employing the presented 3D system and longitudinal imaging pipeline together with appropriate fluorescent-tagged reporter proteins.”.

5) The semantics regarding "organoids" should be addressed: The authors call “organoids”, the in vitro 3D culture they use. Not all 3D spheroids culture can be called “organoids”, it is essential first to examine if the 3D structures grown in culture resemble the original breast tissue (e.g. cell composition, geometry). Why not use the well-defined protocol developed for mouse organoid culturing where both main mammary epithelial cell types can be modelled (luminal and myoepithelial) (Jamieson, Dekkers et al., 2016)? The authors should either prove that their cultures can be called organoids or use the right protocol. It is of great importance as they are claiming they use a stochastic model. Looking at shape and polarity, I presume their organoids are luminal-enriched.

We would like to take the opportunity to briefly describe the original primary mammary 3D system and culture conditions used in our lab (Jechlinger et al., 2009, Genes and Development; Havas et al., 2017, JCI). We have anecdotal evidence for K14/SMA positive cells in this 3D system and, hence, referred to these cultures as “organoids” in the aforementioned publications. These cultures permitted us to isolate hallmarks of tumor progression and residual disease that were successfully mapped to tissues and expression data of breast cancer patients following neoadjuvant therapy (harboring residual disease) (please refer to Havas et al., 2017 and Radic Shechter et al., 2020, bioRxiv). Building on these experiences and data we decided to stay with these conditions for correlation with a stochastic model (please see also Figure 1—figure supplement 1).

However, we are aware that new culture conditions allow for better compartmentalization of cell types within 3D organoid cultures. Since our mammary epithelial cell cultures are – in fact as the reviewer points out – predominantly luminal epithelial, we accede to rename our spheroids “luminal epithelial acini”/ simply “acini”. The terminology has been changed everywhere in the manuscript.

6) Studies regarding polarity should be strengthened: 3D z-stack and rendering should have been performed to make conclusive interpretation in Figure 2A about disrupted polarity and random lumen. 2D often lead to a wrong interpretation of architectural phenotypes. Moreover, the authors should expand the polarity analyses by measuring a number of polarity markers (ZO-1, Par3).

Despite the unfortunate work situation, we feel that we could address all essential revision points with the exception of this one. However, since we are already showing a 2D analysis on alpha6-integrin and ZO1 (Figure 2A) and considering the currently insecure process of obtaining reagents as well as restrictions on accessing the microscopy facility, we were hoping for your permission to forego this revision point. Having said this, EMBL has only recently begun to gradually re-open for pressing experiments and COVID19 related research and we will try our best to perform the immunofluorescence analysis as fast as possible, in case you deem this still necessary.

7) The authors should refine terminology related to localized transformation and/or perform the experiments that would allow them to suggest that transformation is in fact localized: The authors also claimed a model of localized transformation while what they did is random and low rate transformation using a doxycycline inducible model with lentiviral delivery of rtTA. It is a confusing claim as localized transformation implies you control the location. This can be performed with optogenetic technology. Alternatively, the terminology used should be refined.

We have removed this misleading terminology and understand that it could be interpreted wrongly.

8) Please ensure the supplementary figures are uploaded.

We have made sure that all figures (now 4 main figures and 6 supplementary figures) are present in the revised manuscript file (PDF version) and also sent as single PDFs as suggested in the guidelines.

9) The authors should not suggest that a big-data analysis was conducted and/or should apply more robust analytical techniques: Authors claim that they do big-data analysis, however only 20 organoids are segmented and tracked. The authors should either provide a more robust analysis or change the semantics that are used in the paper.

We agree that “big-data” analysis can be misleading. What we were trying to convey, is that the imaging data obtained during longitudinal SPIM imaging is fairly large in size and difficult to handle (more than a terabyte per acinus, Figure 3—figure supplement 2). We have substituted the misleading terminology with “big-image-data” to avoid confusion.

Reviewer #1:[…]As indicated by the authors, single cell RNA sequencing should be done to determine how cells in clusters differ from those which are more isolated. Alternatively, key factors may be analyzed perhaps with IHC, to determine if there are tangible differences.

Regarding single cell RNA sequencing, we have mentioned this possibility in the Discussion as an outlook and write above in response “(We) agree with the suggested revision points and have aimed to temper the claims as well as focused on technical revisions.”

As for IF analyses, we have to refer to our answer outlined to the Essential revisions point 6.

Reviewer #2:[…]2) What impact would co-cultivation of myoepithelial cells have on the observed tumor dynamics.

As we outlined above in the Essential revisions point 5, we have anecdotal evidence of myoepithelial cells in these culture conditions. Due to their sparseness, it is highly unlikely that cells are infected in the stochastic system and contribute to clonal expansion. In fact, we rely on this system for an easily accessible luminal epithelial layer for the viral transduction. The presence of myoepithelial cell enriched media or culture conditions that foster their outgrowth after the transduction step, represents a future aim to further explore tumor initiation within an otherwise healthy mammary organoid.

Reviewer #3:[…]– Authors claim that they do big-data analysis, however only 20 organoids are segmented and tracked, this is misleading. The authors should either provide a more robust analysis or change the semantics that are used in the paper. Also, few terabytes per organoid imaged…I may advise to consider other imaging technologies such as confocal that will give better images for less data generated.

Please refer to our answer outlined to the Essential revisions point 9.

Regarding the suggestion of employing confocal microscopy: This has been tried and was clearly resulting in phototoxic effects when imaging was carried out for even a few hours at 10-15 minute intervals (to not miss out on the directionality of cell division events which necessary to perform single cell tracing). Spinning disk confocal was less phototoxic but did not match the image clarity results obtained from light sheet microscopy (SPIM). Furthermore, resolution of images 2-3 cell-layers from the outer rim of the acinus towards the center was largely reduced with confocal technology as compared to light sheet imaging using low (non-phototoxic) laser doses.